# FineBench: Benchmarking and Enhancing Vision-Language Models for Fine-grained Human Activity Understanding

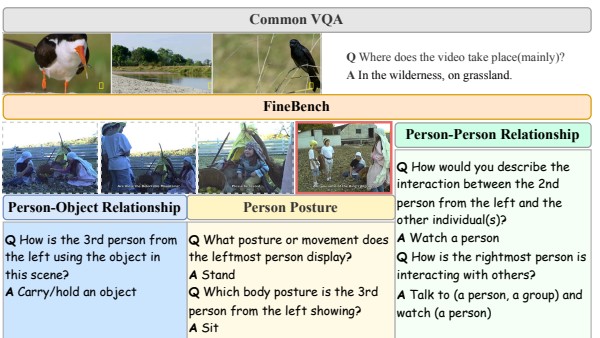 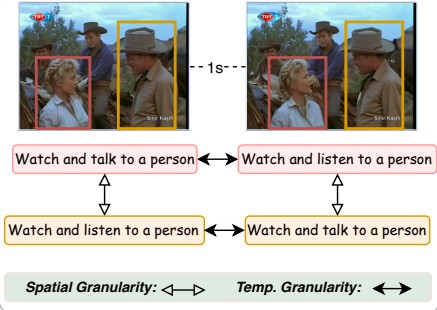

(a) FineBench versus regular VQAs  (b) FineBench's Levels of Granularity

Figure 1: **(a)** Examples of question types in FineBench which go beyond summarization to cover person posture, person-object interactions, and person-person interactions. **(b)** The capture of temporal evolution of interaction labels across frames, emphasizing spatial granularity (e.g., distinguishing among individuals in the same frame) and temporal granularity (e.g., resolving transitions between similar but distinct actions).

## Abstract

Vision-Language Models (VLMs) have demonstrated remarkable capabilities in general video understanding, yet they often struggle with the fine-grained comprehension crucial for real-world applications requiring nuanced interpretation of human actions and interactions. While some recent human-centric benchmarks evaluate aspects of model behaviour such as fairness/ethics, emotion perception, and broader human-centric metrics, they do not combine long-form videos, very dense QA coverage, and frame-level spatial/temporal grounding at scale. To bridge this gap, we introduce FineBench, a human-centric video question answering (VQA) benchmark specifically designed to assess fine-grained[1] understanding. FineBench comprises 199,420 multiple-choice QA pairs densely annotated across 64 long-form videos (15 minutes each), focusing on detailed person movement, person interaction, and object manipulation, including compositional actions. Our extensive evaluation reveals that while proprietary models like GPT-5 achieve respectable performance, current open-source VLMs significantly underperform, struggling particularly with spatial reasoning in multi-person scenes and distinguishing subtle differences in human movements and interactions. To address these identified weaknesses, we propose FineAgent, a modular framework that enhances VLMs by leveraging a Localizer and a Descriptor. Experiments show that FineAgent consistently improves the performance of various open VLMs on FineBench. FineBench provides a rigorous testbed for future research into fine-grained human-centric video understanding, while FineAgent offers a practical approach to enhance such reasoning in current VLMs.

---

[1]See Appendix A.1 for our definition of "fine-grained" in this paper.

# 1 INTRODUCTION

Vision-Language Models (VLMs) are rapidly advancing, showing increasing proficiency in interpreting and reasoning about visual content, particularly in the domain of video understanding. However, much of the focus has been on general comprehension tasks—recognizing overall scenes, identifying high-level activities, or summarizing broad events. While valuable, this often falls short in real-world scenarios demanding a *fine-grained* understanding of video content involving humans. Fine-grained video understanding requires perceiving subtle visual details, precise temporal dynamics of actions, complex spatial relationships, and nuanced interactions, especially concerning human behavior. For instance, distinguishing between a person deliberately sitting versus accidentally falling, or discerning intricate social cues in a conversation, requires a level of detail beyond general scene description. Such capabilities are critical for applications ranging from assistive technologies and healthcare monitoring to autonomous systems and detailed behavior analysis.

Despite its importance, fine-grained, human-centric video understanding remains relatively under-explored and under-evaluated in the current VLM landscape. Existing Video Question Answering (VQA) benchmarks often rely on sparsely annotated clips, focus on object-centric or broad activity recognition, or lack the scale and density needed to probe deep, temporally-grounded comprehension (Yu et al., 2019; Xu et al., 2016; Xiao et al., 2021; Li et al., 2024b). As highlighted in Table 1, existing benchmarks often lack a specific focus on fine-grained human-centric actions, dense temporal and spatial grounding, or the sheer density of questions required to thoroughly test nuanced reasoning over extended video durations. This gap hinders progress, as we lack standardized ways to measure and drive improvements in VLMs' ability to grasp subtle human behavior in videos.

To address this gap, we introduce **FineBench**, a new benchmark specifically designed to evaluate fine-grained, human-centric video understanding. FineBench is formulated as a multiple-choice VQA dataset containing nearly 200,000 QA pairs derived from 64 long-form videos. Uniquely, it features dense annotations, averaging over 3,100 questions and linking to approximately 785 distinct keyframes per video, enabling detailed assessment of model capabilities at a granular temporal level (e.g., seconds). The questions cover three core domains: Person Movement, Person Interaction, and Object Manipulation, with over 20% requiring compositional reasoning about combined actions. FineBench explicitly tests spatial and temporal precision through carefully constructed questions and distractors derived from the rich annotations of the AVA v2.2 dataset (Gu et al., 2018).

Using FineBench, we conduct a comprehensive evaluation of state-of-the-art VLMs, encompassing both leading proprietary models (e.g., GPT-4o, Gemini-1.5-Flash, Gemini-2.0-Flash) and a wide array of open-source models (e.g., InternVL-2.5, Qwen-VL2.5, MiniCPM-2.6). Our findings, detailed in Section 3.3 and summarized in Table 3, reveal a significant performance disparity. While powerful proprietary models demonstrate strong capabilities, achieving accuracies above 70% on a representative subset, open-source models struggle considerably. Further analysis (Section 3.4, Figure 3) pinpoints specific weaknesses: VLMs exhibit a marked decline in accuracy as the number of people in the scene increases, indicating challenges with spatial reasoning and subject disambiguation. Furthermore, the tested models consistently perform worse on nuanced Person Movement and Person Interaction tasks compared to more visually distinct Object Manipulation actions.

Motivated by these findings, we propose **FineAgent**, a modular framework designed to enhance the fine-grained video understanding capabilities of existing VLMs by directly addressing the identified bottlenecks (Section 4). FineAgent integrates two key components: a *Localizer* that provides explicit bounding box information to aid subject disambiguation in complex scenes, and a *Descriptor* that generates frame summaries, thereby providing richer semantic context.

Our main contributions are as follows:

- We introduce FineBench, the first densely annotated, human-centric VQA benchmark specifically targeting fine-grained video understanding, featuring 199,420 QA pairs.
- We provide a comprehensive benchmark of current proprietary and open-source VLMs on FineBench, revealing significant limitations in open-source models' fine-grained reasoning abilities, particularly in spatial reasoning and nuanced action interpretation.
- We conduct an in-depth analysis identifying key failure modes for VLMs: degraded performance in multi-person scenarios (spatial reasoning) and difficulties understanding nuanced human movements and interactions.

- We propose FineAgent, a modular framework leveraging spatial grounding and contextual captioning, demonstrating its effectiveness in improving the fine-grained video understanding performance of various open-source VLMs by targeting their specific weaknesses.

Table 1: Comparison of FineBench with existing VQA datasets across key dimensions. Our dataset is the first to combine fine-grained actions, dense temporal grounding (Temporal G.), dense spatial grounding (Spatial G.), and large-scale multiple-choice QA in a human-centric setting.

| | Num. QAs | Num. Videos | Avg. Duration (s) | Density | Human-Centric | Spatial G. | Temporal G. |
|---|---|---|---|---|---|---|---|
| VideoMME (Fu et al., 2024) | 2,700 | 900 | 1017.9 | 3 | ✗ | ✗ | ✗ |
| EgoSchema (Mangalam et al., 2023) | 5,063 | 5,063 | – | 1 | ✓ | ✗ | ✗ |
| MovieChat-1k (Song et al., 2024) | 13,000 | 1000 | 564 | 13 | ✗ | ✗ | ✗ |
| ActivityNet-QA (Yu et al., 2019) | 8,000 | 800 | 111.4 | 10 | ✗ | ✗ | Partial |
| LongVideoBench (Wu et al., 2024b) | 6,678 | 3,763 | 473 | 1.8 | ✗ | ✗ | Partial |
| NExT-QA (Xiao et al., 2021) | 8,564 | 1,000 | 39.5 | 8.6 | ✗ | ✗ | ✗ |
| MSRVTT-QA (Xu et al., 2016) | 72,821 | 2,990 | 15.2 | 24.4 | ✗ | ✗ | ✗ |
| MSVD-QA (Xu et al., 2016) | 13,157 | 504 | 9.8 | 26.1 | ✗ | ✗ | ✗ |
| STAR (Wu et al., 2024a) | 7,098 | 914 | 11.9 | 7.8 | ✗ | ✗ | ✗ |
| MVBench (Li et al., 2024b) | 4,000 | 3,641 | 16 | 1.1 | ✗ | ✗ | ✗ |
| TemporalBench (Cai et al., 2024) | 10,000 | 2000 | – | 5 | ✗ | ✗ | Partial |
| HV-MMBench (Cai et al., 2025) | 8,700 | 1,200 | – | 7.25 | ✓ | ✗ | ✗ |
| **FineBench (Ours)** | 199,420 | 64 | 900 | 3115.94 | ✓ | ✓ | ✓ |

## 2 RELATED WORK

Our work on FineBench builds upon extensive research in Video Question Answering (VQA) and the rapid advancements in Vision-Language Models (VLMs).

**Video Question Answering Datasets**. VQA evaluates video understanding via question answering. While numerous datasets exist, early influential ones like MSRVTT-QA (Xu et al., 2016) and ActivityNet-QA (Yu et al., 2019) often lacked dense spatial or temporal grounding, limiting fine-grained evaluation (Table 1). Subsequent datasets focused on deeper reasoning (e.g., NExT-QA (Xiao et al., 2021), STAR (Wu et al., 2024a)) or specialized domains like egocentric video (EgoSchema (Mangalam et al., 2023)). Recent benchmarks (e.g., MovieChat (Song et al., 2024), MVBench (Li et al., 2024b), TemporalBench (Cai et al., 2024), and MovieCORE (Faure et al., 2025)) address various aspects like long videos or temporal reasoning. Some benchmarks explicitly emphasize human-centric evaluation. HumaniBench (Raza et al., 2025) focuses on human-centered AI principles such as fairness and empathy through image tasks, whereas HumanVBench (Zhou et al., 2024) explores human-centric video understanding with synthetic data pipelines targeting emotion perception and speech–visual alignment. However, a gap remains for evaluating fine-grained human action understanding with dense grounding, particularly in complex scenes. FineBench addresses this gap by providing large-scale QA with dense *spatial and temporal grounding of human actions and interactions* in long videos (avg. 900s), facilitating rigorous evaluation of precise human behavior localization and comprehension.

**Vision-Language Models (VLMs)**. Vision-Language Models (VLMs), integrating vision encoders and LLMs, have revolutionized cross-modal understanding with early works such as LlaVA (Liu et al., 2023), MiniCPM-v2.6 (Yao et al., 2024), and more recently, InternVL-2.5 (Chen et al., 2024) and Qwen2.5-VL (Bai et al., 2025). Extending this to video, recent VLMs like, mPlugOwl-3 (Ye et al., 2024a), and HERMES (Faure et al., 2024) handle temporal information to perform video tasks, including video captioning and VQA. Despite their capabilities, our analysis (Section 3.3 and Section 3.4) reveals significant challenges for these models in fine-grained video understanding, particularly concerning spatial localization in complex scenes and interpreting nuanced human actions and interactions. This underscores the need for human-centric benchmarks like FineBench.

## 3 FINEBENCH

To effectively evaluate Vision-Language Models' (VLMs) capacity for understanding nuanced visual content, we first delineate the characteristics of fine-grained video understanding as distinct from the general video understanding typically assessed by existing VQA datasets (Section 3.1) and overview FineBench. Section 3.2 then elaborates on our data creation and annotation process.

Subsequently, we present extensive experiments benchmarking current VLMs to assess their proficiency in fine-grained video comprehension (Section 3.3). Finally, Section 3.4 examines the primary reasons these models struggle with such a task, providing insights for performance enhancements.

Table 2: Key Statistics of FineBench.

| Statistic | Value |
|---|---|
| Total Questions | 199,420 |
| Unique Videos | 64 |
| Avg. Annotated Frames/Video | 785 |
| *Category Distribution:* | |
| Person Movement | 94,330 (47.30%) |
| Person Interaction | 70,140 (35.17%) |
| Object Manipulation | 34,950 (17.53%) |
| *Question Composition:* | |
| Single Actions | 158,625 (79.54%) |
| Combined Actions | 40,795 (20.46%) |

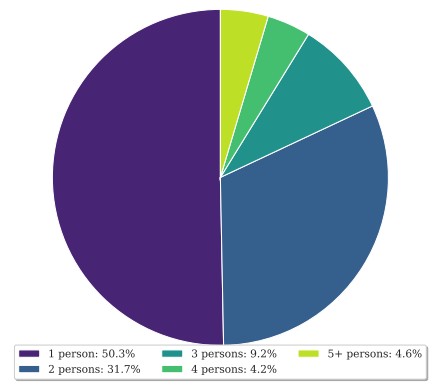

Figure 2: Distribution of Annotated Persons per Keyframe.

## 3.1 OVERVIEW OF FINEBENCH

**Fine-grained video understanding** represents a crucial yet relatively underexplored facet of video-language models (VLMs). Unlike general video understanding tasks that focus on broad concepts, scene recognition, or high-level activities, fine-grained understanding requires models to perceive and reason about subtle visual details, momentary actions, and precise object interactions within video frames. A fine-grained *human-centric* VQA dataset, in particular, must offer comprehensive coverage of all observable human behaviors. This includes not only the subject's body pose and movement, but also their interactions with objects (*person-object interactions*) and with other individuals (*person-person interactions*). Figure 1a illustrates this diversity by showcasing QA examples across different reasoning types supported by FineBench, from posture recognition to complex social interactions. Figure 1b highlights the temporal and spatial granularity required, where action labels evolve across frames and demand fine discrimination between visually similar behaviors.

**A "Fine-grained" video understanding system** must possess the ability to distinguish between visually similar activities that share common attributes. In our context, this includes precise action disambiguation (between similar actions such as "carrying" versus "lifting" an object), temporal precision (identifying when actions start and end), spatial attention (focusing on the relevant regions within a frame), and contextual reasoning (understanding actions in relation to the environment).

**The importance of fine-grained understanding for VLMs** becomes evident when considering practical applications. In surveillance scenarios, general understanding might merely identify "several people in a room," whereas fine-grained perception can distinguish whether individuals are "standing in conversation," "reaching for objects," or "exhibiting suspicious behavior." For assisted living monitoring, fine-grained understanding allows systems to differentiate between "a person deliberately sitting down" versus "a person losing balance and falling"—a critical distinction for emergency response. Similar examples exist across sports analysis and everyday activities, positioning fine-grained video understanding as a fundamental capability that VLMs must possess to function effectively in complex real-world scenarios where subtle distinctions carry significant meaning.

**Our human-centric fine-grained video benchmark, FineBench**, is structured as a multiple-choice video question answering (VQA) dataset, where each question is accompanied by four candidate answers, only one of which is correct. It contains a total of **199,420 QA pairs**, making it one of the largest VQA datasets. Questions are densely linked to an average of **785 unique keyframes** per video across **64 long-form videos**, enabling detailed probing of model understanding at the second level. Unlike existing VQA datasets that focus on general comprehension or sparse annotation, FineBench offers an average of **3,100 QA pairs per video**, fostering local and holistic reasoning.

Table 2 summarizes the key statistics of FineBench. The dataset spans three broad conceptual domains—*movement*, *human interaction*, and *object manipulation*—which guide the diversity of visual reasoning required. **over 20% of QA pairs involve combined activities**, testing compositional reasoning where multiple visual cues must be integrated to answer correctly. Figure 2 shows that nearly half the frames contain multiple annotated persons, emphasizing the fine-grained nature of the interactions present in FineBench. These properties (along with those in Table 1) make FineBench the first benchmark explicitly designed to test VLMs' fine-grained human-centric video understanding ability, where success requires precision in space, time, and context.

## 3.2 DATASET CREATION PROCESS

The construction of FineBench leverages the human-annotated action classes and bounding boxes provided by the AVA dataset (Gu et al., 2018). Our methodology integrates three core components: (1) systematic question generation using predefined templates, (2) a principled distractor selection strategy, and (3) spatial reasoning for subject disambiguation and subject-specific QA generation.

### 3.2.1 QUESTION TEMPLATE DESIGN AND INSTANTIATION

We design a structured set of question templates categorized by the nature of the action being queried. Specifically, 23 templates were created for *person movement* actions (e.g., "How would you describe the movement of {*person*}?"), 21 templates for *object manipulation* actions (e.g., "How is {*person*} interacting with the object?"), and 25 templates for *person interaction* actions (e.g., "What social interaction is {*person*} engaged in?"). To anchor these questions visually and ensure clarity, the placeholder {*person*} within each template is instantiated using spatial descriptors derived dynamically from bounding box positions. Phrases such as "the leftmost person" or "the person in the center" are employed to unambiguously refer to the specific individual relevant to the question within the video frame. (See Appendix A.5 for the full lists of templates).

### 3.2.2 DISTRACTOR GENERATION STRATEGY

For each annotated action instance in AVA v2.2, we generate a corresponding multiple-choice question. The process begins by classifying the ground truth action into one of the three categories: person movement, object manipulation, or person interaction. A question template is then randomly selected from the pool corresponding to that action category. Plausible distractors (incorrect answer options) are generated using a two-tiered approach. The primary strategy involves selecting actions that are semantically similar to the correct answer, based on a predefined similarity mapping. For example, actions like "hand wave", "hand clap", and "hand shake" are considered semantically close and may serve as distractors for one another, thereby increasing the question's difficulty. If no sufficiently similar actions are found via this mapping, a fallback strategy is employed: distractors are randomly selected from the same broad action category (e.g., other person movement actions) to maintain contextual relevance. In scenarios where an individual is annotated with multiple concurrent actions belonging to the same category, we formulate compound questions (e.g., reflecting simultaneous actions like "listening to and watching a person") and select appropriate distractors.

### 3.2.3 SPATIAL REFERENCING AND DISAMBIGUATION

To enable precise questioning about specific individuals within a scene, especially when multiple people are present, we implement a dynamic spatial referencing system based on bounding box locations. When only one or two individuals are detected, relative positional terms (e.g., "the person on the left", "the person on the right") are used for disambiguation. For scenes containing three or more individuals, ordinal references (e.g., "the second person from the left") are generated to ensure clarity. This ensures that the generated questions unambiguously target the intended person.

## 3.3 DO VLMS EXHIBIT FINE-GRAINED VIDEO UNDERSTANDING?

To evaluate whether current Vision-Language Models (VLMs) can perform fine-grained human-centric video understanding, we benchmark a diverse set of proprietary and open models using FineBench, integrated into the VLMEvalkit (Duan et al., 2024) library. Due to the high cost of

Table 3: **Performance of 15 Vision-Language Models (VLMs) on FineBench**. Proprietary models evaluated on a representative subset–comprising 7 representative videos and totaling 20,143 questions–are shown at the top. Open models are evaluated on both the subset and the full dataset. The best full-dataset open score is **bolded** and the second-best underlined. [P.: Person; Obj.: Object]

| | Size | P. Movement | P. Interaction | Obj. Manipulation | Avg. |
|---|---|---|---|---|---|
| Random Choice | – | 25.0 | 25.0 | 25.0 | 25.0 |
| *Subset Evaluation* | | | | | |
| GPT-4o (2024/08/26) (OpenAI, 2024) | – | 70.9 | 73.9 | 84.4 | 74.3 |
| GPT-5-mini (2025/08/07) (OpenAI, 2025) | – | **75.9** | **75.3** | 85.3 | **77.4** |
| Gemini-1.5-Flash (Team et al., 2024) | – | 71.2 | 66.8 | 81.9 | 71.6 |
| Gemini-2.0-Flash (Team et al., 2024) | – | **75.9** | 68.7 | **86.3** | 75.2 |
| SmolVLM (Marafioti et al., 2025) | 2B | 48.5 | 48.0 | 80.0 | 53.9 |
| MiniCPM-2.6 (Yao et al., 2024) | 8B | 49.5 | 57.4 | 84.8 | 58.4 |
| mPlugOwl-3 (Ye et al., 2024a) | 7B | 47.9 | 55.8 | 84.0 | 56.6 |
| *Full Dataset Evaluation* | | | | | |
| InternVL-2.5 (Chen et al., 2024) | 1B | 33.8 | 40.2 | **79.6** | 44.1 |
| SmolVLM (Marafioti et al., 2025) | 2B | 47.9 | 50.5 | 71.0 | 52.9 |
| Qwen-VL2.5 (Bai et al., 2025) | 3B | 58.0 | 57.5 | 73.2 | 60.5 |
| BLIP-3 (Xue et al., 2024) | 4B | 34.3 | 58.6 | 64.9 | 48.2 |
| InternVL-2.5 (Chen et al., 2024) | 4B | 61.4 | 58.6 | 78.1 | 63.3 |
| mPlugOwl-2 (Ye et al., 2024b) | 7B | 57.6 | 49.2 | 78.5 | 58.3 |
| mPlugOwl-3 (Ye et al., 2024a) | 7B | 48.9 | 54.8 | 75.2 | 55.6 |
| MiniCPM-2.6 (Yao et al., 2024) | 8B | 56.2 | 56.5 | 72.8 | 59.2 |
| LLaVA-OV (Li et al., 2024a) | 7B | 53.3 | 60.4 | 69.6 | 58.6 |
| InternVL-2.5 (Chen et al., 2024) | 8B | 66.8 | 62.1 | 78.1 | 67.1 |
| Qwen-VL2.5 (Bai et al., 2025) | 7B | **70.7** | **63.8** | 73.9 | **68.8** |

querying proprietary APIs at scale, we provide results on two tiers: a representative subset[2] (7 videos, 20,143 QAs) and the full dataset for open models only. The results are shown in Table 3.

Proprietary models, notably GPT-5-mini (OpenAI, 2025) and Gemini-2.0-Flash (Team et al., 2024), demonstrate strong performance on the representative subset, substantially outperforming open models evaluated on the same data (e.g., MiniCPM-2.6 (Yao et al., 2024) at 58.4%). This suggests these models possess stronger spatio-temporal reasoning and fine-grained human activity disambiguation capabilities, likely due to large-scale pretraining and robust multimodal pipelines.

In contrast, open models exhibit wide variability and underwhelming accuracy on the full dataset. The top open model, Qwen-VL2.5 (7B) (Bai et al., 2025), achieves 68.8%, but most models cluster around 55–60%, and a few perform near chance level on Person Movement-related questions. These gaps indicate that current open VLMs struggle with fine-grained temporal cues, subtle interactions, and compositional reasoning—core challenges posed by FineBench. Such results highlight a critical gap in the open ecosystem and a need for progress in training methods, architectures, and benchmarks tailored for fine-grained human-centric video comprehension.

## 3.4 WHY DO VLMs STRUGGLE WITH FINE-GRAINED VIDEO UNDERSTANDING?

Having established that current Vision-Language Models (VLMs) underperform on fine-grained video understanding tasks (Section 3.3), we investigate the underlying reasons by dissecting their performance. Our analysis focuses on two key aspects: the impact of scene complexity (number of persons) and the variation in performance across different action categories, visualized through radar charts in Figure 3a and Figure 3b, respectively. Additionally, we investigate the influence of input context length to ascertain if insufficient visual information is a bottleneck, as shown in Figure 3c.

First, analyzing the accuracy relative to the number of people present (Figure 3a) reveals a significant and consistent challenge for all evaluated VLMs. There is a clear trend of performance degradation as the number of individuals in the frame increases. For example, Qwen-VL2.5 (7B),

---

[2]Same annotations distribution as the full set. See Appendix A.6

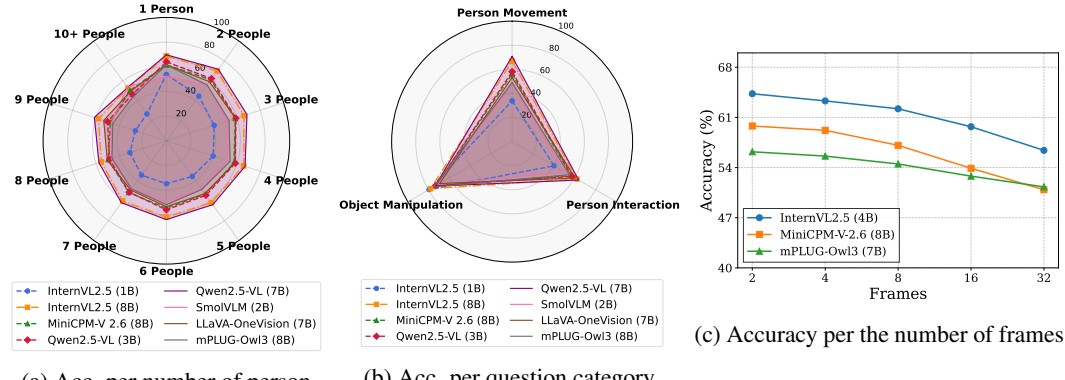

(a) Acc. per number of person     (b) Acc. per question category

(c) Accuracy per the number of frames

Figure 3: VLM performance analysis on FineBench detailing accuracy variations. (a) Performance degradation with increasing number of persons in the scene. (b) Performance differences across action categories, with Person Movement being consistently lower. (c) Performance degradation with increasing number of frames.

the top-performing model overall, has a peak accuracy of 71.7% in scenes with 2 persons, but this accuracy drops to 53.4% when 10 or more people are present. This decline is even more pronounced for smaller models like InternVL-2.5 (1B), which drops from 53.7% to 26.9%. This consistent decrease suggests that VLMs struggle significantly with spatial reasoning, target disambiguation, and relationship understanding in complex, multi-agent scenarios. Identifying and tracking the specific actions of designated individuals becomes substantially harder amidst visual clutter, potential occlusions, and the need to interpret complex spatial references (e.g., "the third person from the left").

Second, examining performance across action categories (Figure 3b) highlights another area of weakness. Models consistently demonstrate higher proficiency in identifying *Object Manipulation* actions compared to *Person Movement* and *Person Interaction*. Across all tested models, accuracy for Object Manipulation typically ranges from 71% to nearly 80%, whereas accuracies for the other two categories are often considerably lower. For instance, InternVL-2.5 (8B) achieves 78.1% on Object Manipulation but only 66.8% on Person Movement and 62.1% on Person Interaction. This disparity suggests that VLMs find it easier to recognize actions centered around distinct object interactions, which may offer clearer visual cues. Conversely, they appear less capable of interpreting the nuances of human kinematics involved in diverse movements and the complex, often subtle, cues defining social interactions between individuals. These person-centric categories demand a deeper understanding of human pose, gestures, and context that current models do not fully capture. We also isolate the impact of the vision components with a blind evaluation of VLMs in Appendix A.3.

Our key takeaway is that current open-source VLMs struggle with fine-grained video understanding primarily due to two challenges. First, they exhibit deficiencies in robust spatial reasoning and subject disambiguation, particularly as scene complexity (number of actors) increases. This makes it difficult to correctly attribute actions to the right individuals. Second, they find it harder to interpret and distinguish nuanced human-centric actions, especially subtle body movements and complex social interactions, compared to more visually salient object-related actions. These person-centered tasks require models to pick up on fine-grained visual details and temporal patterns of human behavior, which current architectures and training paradigms are not yet adept at. Addressing these limitations is key for advancing fine-grained human-centric video understanding.

## 4 FINEAGENT

Our error analysis in Section 3.4 identifies two primary obstacles hindering the fine-grained video understanding capabilities of current Vision-Language Models (VLMs): (1) difficulties with spatial reasoning and subject disambiguation in multi-person scenes, and (2) a weaker grasp of nuanced human movements and interactions compared to object-centric actions. To address these limitations, we propose **FineAgent**, a modular framework designed to augment existing VLMs with spatial grounding and contextual information, thereby enhancing their fine-grained reasoning abilities.

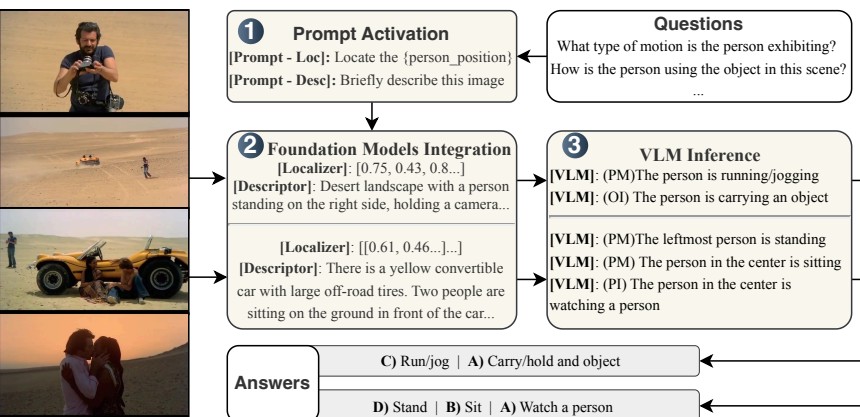

Figure 4: **Workflow of FineAgent**. It begins with (1) prompt activation for the Localizer and Descriptor. (2) The Localizer and Descriptor, both Foundation models, provide bounding box coordinates and textual captions. (3) Finally, the VLM uses this processed information during inference.

### 4.1 HOW DOES FINEAGENT ENHANCES FINE-GRAINED VIDEO UNDERSTANDING?

**FineAgent** enhances VLMs' fine-grained video understanding capabilities at inference time by integrating two complementary modules, designed to provide information that directly addresses the weaknesses identified in Section 3.4. The overall workflow of FineAgent is illustrated in Figure 4.

The first module is the **Localizer**, instantiated using EVFSam (Zhang et al., 2024), a foundation model adept at visual grounding and referring segmentation. Given the video frames and the question, the Localizer provides the spatial location of the individual pertinent to the query. By supplying positional information, this module directly tackles the VLM's observed struggle with spatial reasoning and subject disambiguation in multi-person scenes. The Localizer thus assists the base VLM in anchoring its visual analysis to the correct subject, mitigating confusion in crowded environments.

The second module is the **Descriptor**. This component is responsible for generating captions for the relevant video frames. We utilize Qwen-VL2.5 (7B) (Bai et al., 2025) as the Descriptor, due to its strong performance among open-source VLMs (Table 3). The Descriptor addresses the VLM's weakness in interpreting subtle human-centric actions, particularly those categorized under Person Movement and Person Interaction (Figure 3b). The generated captions provide semantic context and higher-level descriptions of potentially ambiguous activities. This augments the base VLM's understanding beyond raw visual features and aids in the interpretation of complex kinematics or social cues that might otherwise be missed. These two modules operate synergistically: the Localizer first identifies *who* and *where* the question is focused on, and then the Descriptor provides a richer textual interpretation of *what* is happening. This structured, auxiliary information is then combined with the question and video input, and fed to the VLM to facilitate a more informed and accurate prediction.

The effectiveness of integrating **FineAgent** is demonstrated empirically in Table 4 (and qualitatively in Suppl. Section A.7.4). Augmenting various base VLMs—including InternVL-2.5 (1B) (Chen et al., 2024), Qwen-VL2.5 (7B) (Bai et al., 2025), mPLUG-Owl-3 (7B) (Ye et al., 2024a), and MiniCPM-2.6 (8B) (Yao et al., 2024) with **FineAgent** framework consistently yields performance improvements across all models and action categories on FineBench. Notably, the improvements are often most pronounced in the challenging Person Movement and Person Interaction categories, directly addressing the identified weaknesses. For instance, augmenting the InternVL-2.5 (1B) model with **FineAgent** boosts its Person Movement accuracy by a substantial 14.1 percentage points and Person Interaction accuracy by 4.0 points, resulting in an overall 8.3-point increase in average accuracy. Similar positive trends, with varying magnitudes, are observed across the other models [3]. This validates our hypothesis that by specifically targeting spatial grounding and providing richer contextual descriptions for human actions, **FineAgent** can successfully enhance the fine-grained video understanding capabilities of existing VLMs.

---

[3]Less so for Qwen-VL2.5 due to limited complementarity. Please see Appendix A.2 and Appendix Table 6

Table 4: Performance gains with FineAgent across different models.

| Model | P. Movement | P. Interaction | Obj. Manipulation | Avg. |
|---|---|---|---|---|
| InternVL-2.5 (1B) (Chen et al., 2024) | 33.8 | 40.2 | 79.6 | 44.1 |
| + **FineAgent** | **47.9** (+14.1) | **44.2** (+4.0) | **80.6** (+1.0) | **52.4** (+8.3) |
| Qwen-VL2.5 (7B) (Bai et al., 2025) | 70.7 | 63.8 | 73.9 | 68.8 |
| + **FineAgent** | **71.5** (+0.8) | **64.1** (+0.3) | **76.3** (+2.4) | **69.7** (+0.9) |
| mPlugOwl-3 (7B) (Ye et al., 2024a) | 48.9 | 54.8 | 75.2 | 55.6 |
| + **FineAgent** | **60.8** (+11.9) | **57.8** (+3.0) | **77.4** (+2.2) | **62.7** (+7.1) |
| MiniCPM-2.6 (8B) (Yao et al., 2024) | 56.2 | 56.5 | 72.8 | 59.2 |
| + **FineAgent** | **60.6** (+4.4) | **58.8** (+2.3) | **76.3** (+3.5) | **62.7** (+3.5) |

## 4.2 Importance of FineAgent Components: Ablation Study

To understand the individual contributions of the proposed modules within FineAgent, we provide ablations on the Localizer and the Descriptor. The results, compared against the base models, are presented in Table 5. Adding only the Localizer yields accuracy improvements for both base models (+2.8% for mPlugOwl-3, +0.5% for Qwen-VL2.5). Such gain suggests that providing explicit spatial grounding helps the models disambiguate subjects and interpret spatial references, directly addressing the key challenge identified in Section 3.3 concerning performance degradation in multi-person scenes. The contribution of the Descriptor module is particularly significant for mPlugOwl-3, which sees a substantial +6.9% increase in accuracy when augmented solely with contextual captions. The improvement for the Qwen-VL2.5 base model, however, is marginal (+0.7%). This difference is explained by the implementation of our Descriptor module, which itself leverages Qwen-VL2.5 to generate the captions. Consequently, when this module is applied to the Qwen-VL2.5 base VLM, it provides limited additional descriptive power. Conversely, for a different architecture like mPlugOwl-3, the Qwen-VL2.5-based captions offer significant external semantic context, helping in addressing the second major weakness identified.

Finally, employing the complete FineAgent framework, which combines both modules, results in the highest overall performance for both models (+8.1% total gain for mPlugOwl-3, +0.9% for Qwen-VL2.5). The combined improvement exceeds the sum of individual gains, indicating a synergistic effect where explicit localization and description complement each other. These

Table 5: Ablation study on FineAgent components. We report average accuracy (%) on FineBench. Each column corresponds to adding a specific module to the base VLM. Improvements over the base model are shown in green.

| Model | + Localizer | | + Descriptor | | + FineAgent | |
|---|---|---|---|---|---|---|
| | Acc. | Δ | Acc. | Δ | Acc. | Δ |
| mPlugOwl-3 (7B) | 58.4 | (+2.8) | 62.5 | (+6.9) | **63.7** | (+8.1) |
| Qwen-VL2.5 (7B) | 69.3 | (+0.5) | 69.5 | (+0.7) | **69.7** | (+0.9) |

results validate that explicitly addressing spatial grounding challenges and enhancing semantic context for actions are crucial strategies for improving fine-grained video understanding in VLMs.

## 5 Conclusion

This work addresses the challenge of fine-grained, human-centric video understanding in Vision-Language Models (VLMs). We introduce FineBench, a densely annotated VQA benchmark with 199 420 question-answer pairs specifically designed to evaluate these nuanced capabilities. Our evaluations reveal that while proprietary models perform well, open-source VLMs struggle significantly, particularly with spatial reasoning in multi-person scenes and distinguishing subtle human movements and interactions. To mitigate these issues, we propose FineAgent, a modular framework incorporating spatial grounding and contextual captioning. Experiments demonstrate that FineAgent consistently improves the performance of various open VLMs on FineBench, especially in the identified areas of weakness. FineBench provides a rigorous testbed for future research, and FineAgent offers a practical approach to enhance the fine-grained reasoning abilities of current models.

## 6 ETHICS STATEMENT

FineBench was developed with careful consideration of ethical aspects in dataset creation and evaluation. Videos are used in accordance with appropriate licenses, and annotations should be used strictly for research purposes only. The benchmark is designed to evaluate fine-grained human action and interaction understanding, not to classify or infer sensitive personal attributes such as race, gender, or emotion. The dataset is released solely for academic research with the intention of advancing robust multimodal understanding.

## 7 REPRODUCIBILITY STATEMENT

To ensure the reproducibility of our work and spur adoption of our benchmark, we will release the dataset as a HuggingFace repository upon acceptance. We provide the code to reproduce our experiments using the VLMEvalKit Duan et al. (2024) library on GitHub with explanations and a ready-made script to reproduce our experiments and run inference with any VLM supported in VLMEvalKit at this anonymous repository `https://anonymous.4open.science/r/FineBench_eval-4AD4`. We are also creating a leaderboard for FineBench and preparing a pull request to the official GitHub repository of VLMEvalkit after the review process is complete.

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

# A APPENDIX

## A.1 DEFINITION OF "FINE-GRAINED" IN FINEBENCH

The term *fine-grained* is often used broadly in the vision-language literature to describe detailed or nuanced understanding. In this paper, we adopt a more specific meaning tailored to long-form video understanding. Concretely, *fine-grained* in FineBench refers to:

- **Spatial grounding**: questions are anchored to specific regions, objects, and persons within frames, enabling evaluation of models' ability to localize visual evidence rather than rely on global scene cues.
- **Temporal grounding**: questions reference precise temporal segments, sometimes spanning only a few frames, and require correct identification of when an action or interaction occurs within long videos (average length ∼900s).
- **Human actions and interactions**: our QA pairs emphasize detailed human-centric phenomena such as person movements, person–person interactions, and person–object manipulations, going beyond coarse activity labels.
- **Compositional reasoning**: many questions require integrating multiple spatial or temporal cues (e.g., tracking an object across shots, or reasoning over sequential actions) to arrive at the correct answer.

Thus, "fine-grained" in FineBench does not imply exhaustive coverage of all possible aspects of granularity (e.g., attribute-level object categorization or subtle affective states), but specifically denotes the dense spatial and temporal grounding of human actions and interactions across extended video contexts. This operational definition aligns with the design choices of our dataset and clarifies the scope of our claims.

## A.2 FAQS–QUESTIONS THE AUTHORS HAVE BEEN ASKING EACH OTHER

**Why don't you evaluate very large open-source models (e.g., 72B)?** Running evaluations on such massive models requires substantial computational resources: GPUs, time, and ultimately money. Our goal was not to exhaustively benchmark every possible model size, but to identify meaningful trends across a representative spectrum of models. To balance comprehensiveness and feasibility, we included some of the strongest proprietary models available (e.g., GPT-4o, Gemini), which provide a useful reference point for what is achievable at scale. This strategy allows us to highlight the gap between cutting-edge proprietary systems and current open-source alternatives without incurring prohibitive costs.

**How do you address possible data contamination?** This is an important concern, especially since we leverage AVA (validation set) videos as a source. Unless one records new footage or secures exclusive copyright directly from producers, there is no way to guarantee that a given video has not been partially seen during pretraining. This is a limitation shared by virtually all modern benchmarks. However, our *annotations* are entirely original: we designed fine-grained, human-centric questions that did not exist before. More importantly, we are the first to explore dense, temporally precise VQA grounded in AVA at this level of detail. Thus, even if models were exposed to raw video data, the benchmark still tests capabilities that go far beyond memorization.

**Why focus on fine-grained video understanding instead of broad, high-level comprehension?** High-level scene understanding such as recognizing that "a person is cooking" is relatively mature in current VLMs. However, applications in assistive technologies, safety-critical systems, and human-computer interaction demand much finer distinctions: Did the person *deliberately* sit down, or did they *fall*? These nuanced judgments require compositional reasoning over subtle motion cues and temporal dynamics, which existing benchmarks largely ignore. Our benchmark is designed to push VLMs toward that more challenging and impactful frontier.

**How is this benchmark different from existing video QA datasets?** Prior benchmarks typically (i) rely on short clips, (ii) ask a small number of relatively coarse questions, or (iii) emphasize object-centric rather than human-centric reasoning. In contrast, FineBench provides nearly 200k QA pairs

over long-form videos, with dense temporal coverage (over 3,100 questions per video). This density and human-centric focus uniquely enable rigorous stress-testing of VLMs' ability to reason about fine-grained human actions and interactions.

**Can this dataset be used for training as well as evaluation?**  We design FineBench primarily as a benchmark, however the same strategy can be used to prepare the training data. In the future we will use a subset of that to create a dataset for few-shot finetuning.

**What future directions does this benchmark enable?**  We see three main avenues: (i) developing specialized modules (like our FineAgent) to boost fine-grained reasoning in existing VLMs, (ii) improving temporal and spatial grounding techniques for multi-person settings, and (iii) fostering new datasets or synthetic approaches that complement our human-centric design. We hope FineBench serves as a catalyst for these next steps.

**Why does Qwen-VL2.5 benefit less from FineAgent?**  As shown in Table 4, Qwen-VL2.5 (7B) exhibits only modest improvements when enhanced with FineAgent, especially compared to other open-source baselines. A natural question is whether this reflects a limitation of our framework or of Qwen-VL2.5 itself. To investigate, we conduct an ablation study (Table 6) where we replace Qwen-VL2.5 as the Descriptor with a different VLM (InternVL2.5-8B).

The results reveal that Qwen-VL2.5 indeed benefits more when the Descriptor comes from a distinct model. Specifically, while the base Qwen-VL2.5 improves by only +0.7 to +0.9 points when serving as its own Descriptor, substituting InternVL2.5 as the Descriptor yields larger gains of up to +1.4 points. This suggests that Qwen-VL2.5's weaker improvements are not due to FineAgent being ineffective, but rather due to limited complementarity when Qwen-VL2.5 plays both roles simultaneously. In other words, FineAgent is most effective when the Localizer and Descriptor modules introduce genuinely new information beyond what the base model already encodes.

Table 6: Ablation study using InternVL2.5 (8B) as Descriptor for FineBench inference with Qwen-VL2.5 (7B). We report average accuracy (%) on FineBench. Each column corresponds to adding a specific module to the base VLM. Improvements over the base model are shown in green.

| Model | + Localizer | | + Descriptor | | + FineAgent | |
|---|---|---|---|---|---|---|
| | Acc. | Δ | Acc. | Δ | Acc. | Δ |
| Qwen-VL2.5 (7B) | 69.3 | (+0.5) | 69.5 | (+0.7) | 69.7 | (+0.9) |
| Qwen-VL2.5 (7B) † | 69.3 | (+0.5) | **69.9** | (+1.1) | **70.2** | (+1.4) |

### A.3 Blind Evaluation of VLMs

To assess the contribution of visual input, we compare VLM's performance in its visual-aware mode against a blind version where only the textual (e.g., question) input is provided. As shown in Table 7, all models experience a substantial performance drop, highlighting the critical role of visual understanding in our benchmark.

| Model | Visual-aware | Blind |
|---|---|---|
| Qwen2.5VL (7B) | 68.8 | 43.5 |
| MiniCPM-v2.6 (8B) | 59.2 | 29.9 |
| InternVL-2.5 (8B) | 67.1 | 33.0 |

Table 7: Performance (%) of VLMs in visual-aware vs. blind (LLM-only) settings.

### A.4 Experimental Settings

Most of our experiments are conducted on 8 V100 GPUs (32GB each). Inference time vary widely across models with mPlug-Owl2 Ye et al. (2024b) and mPlug-Owl3 Ye et al. (2024a) taking around

2 hours, MiniCPM-V2.6 Yao et al. (2024) taking 4 to 5 hours and InternVL2.5 IntenVL (2024) (7B) taking a little less than 15 hours per inference cycle. Trading off between enough context and computational complexity, we feed 5 frames to each model per question where the middle frame contains the main frame the question targets.

## A.5 DATA CREATION BLUEPRINT

In this section, we aim to give the reader a clear blueprint of how we create the dataset. The construction of our action recognition dataset employs a systematic methodology centered around the generation of Multiple Choice Questions (MCQs) derived from video action annotations. This blueprint outlines the core principles and stages of our data creation process, emphasizing the design choices made to ensure the dataset's quality and suitability for advancing research in action understanding.

**Leveraging Structured Video Action Annotations**   The foundation of our dataset lies in a collection of video sequences meticulously annotated with human actions. These annotations provide rich information, including temporal context, individual identities via bounding boxes, and corresponding action labels categorized into distinct types such as human movement, object interaction, and interpersonal engagement. This structured annotation framework serves as the primary source from which our question-answering pairs are generated.

**Template-Guided Question Generation**   To transform the action annotations into a queryable format, we utilize a carefully curated set of question templates, stratified by action category. As exemplified with comprehensive listings in Figures 5b, 5a, and 6a. These templates offer a standardized yet flexible approach to formulating questions about the observed actions. Furthermore, the spatial context of the actors within the video frame is considered to create more specific and grounded queries, enhancing the relevance of the generated questions.

**Strategic Design of Distractor Options**   A key contribution of our data creation process is the deliberate strategy employed for generating incorrect answer options (distractors). Recognizing the importance of challenging yet plausible alternatives, our methodology prioritizes semantic relatedness. We utilize a knowledge resource of similar actions (illustrated in Table 9) to generate distractors that closely resemble the correct action, thereby demanding a finer-grained understanding of the visual content. When semantically similar options are limited, the system draws from other actions within the same category and, as a final measure, from the broader action vocabulary, ensuring a comprehensive set of choices for each question.

**Addressing Complex Action Scenarios**   Our blueprint also accounts for instances where individuals perform multiple concurrent or related actions within a single video frame. In such cases, our methodology aggregates these actions into a single, more complex question. The corresponding correct answer reflects the combination of these actions. The distractors for these multi-action questions are designed to maintain a similar level of complexity, often involving combinations of other actions or partial overlaps with the correct set, thus promoting a deeper level of reasoning about the observed activities.

**Ensuring Scalability and Reliability**   The entire data generation pipeline is engineered for scalability, allowing for the efficient creation of a large-scale dataset from extensive video annotations. Furthermore, this rigid template approach contributes to the integrity of the dataset.

## A.6 THE SUBSET

To facilitate more accessible experimentation for researchers with limited resources (ourselves included) we provide a smaller subset of FineBench. This subset is designed to enable rapid prototyping and low-cost benchmarking before scaling up to the full dataset. It consists of 20,143 QA pairs across 7 videos, amounting to approximately 10% of the full dataset. As shown in Table 8, the subset closely mirrors the full dataset (shown in the main paper in Table 2) in terms of category distribution and question composition. Furthermore, as evidenced by our results in Table 3, models evaluated on the subset exhibit similar performance trends to those evaluated on the full dataset.

(a) Object Manipulation Templates

(b) Person Movement Templates

Figure 5: QA templates for Object Manipulation and Person Movement

(a) Person Interaction Templates

Table 8: Key Statistics of FineBench Sub.

| Statistic | Value |
| --- | --- |
| Total Questions | 20,143 |
| Unique Videos | 7 |
| Annotated frames/Video | 806 |
| *Category Distribution:* | |
| Person Movement | 9645 (47.88%) |
| Person Interaction | 6946 (34.48%) |
| Object Manipulation | 3552 (17.63%) |
| *Question Composition:* | |
| Single Actions | 16,709 (82.95%) |
| Combined Actions | 3434 (17.05%) |

## A.7 DETAILS OF FINEAGENT

### A.7.1 THE LOCALIZER

The **Localizer** module in FineAgent is responsible for spatially grounding the entity referenced in a natural language question. Given a frame from the video and the question (e.g., "What is the person on the left doing?"), the Localizer identifies the region in the image corresponding to the referent (e.g., "the person on the left") and produces a segmentation mask and pseudo-bounding box. This localized region is then passed to the VLM at inference.

To achieve this, we leverage the vision-language segmentation model, EVF-SAM Zhang et al. (2024), which supports referring expression segmentation using foundation models. The input question is first parsed using rule-based patterns to extract referential phrases. These phrases guide the segmentation model, which combines features from a BEiT-3 Wang et al. (2023) language-image encoder and a Segment Anything Model (SAM) Kirillov et al. (2023) backbone.

The Localizer outputs a binary segmentation mask and its corresponding bounding box, both of which are used to crop or mask the input image for further reasoning. If no valid region is found (e.g., due to ambiguous reference or occlusion), the system falls back to global reasoning using the entire frame. This spatial grounding module allows FineAgent to dynamically adjust its perceptual field based on linguistic cues, improving fine-grained video understanding accuracy.

Table 9: Examples of Similar Action Categories

| Action | Similar Actions |
|---|---|
| bend/bow (at the waist) | walk, crouch/kneel, sit |
| dance | walk, run/jog, jump/leap |
| fall down | lie/sleep, get up, crouch/kneel |
| lie/sleep | sit, fall down, crouch/kneel |
| run/jog | walk, jump/leap, stand |
| carry/hold (an object) | lift/pick up, put down, throw |
| catch (an object) | throw, carry/hold (an object), lift/pick up |
| chop | cut, cook, stir |
| climb (e.g., a mountain) | walk, run/jog, jump/leap |
| close (e.g., a door, a box) | open (e.g., a window, a car door), pull (an object), push (an object) |
| eat | drink, cook, smoke |
| enter | exit, open (e.g., a window, a car door), close (e.g., a door, a box) |
| hit (an object) | kick (an object), throw, touch (an object) |
| lift/pick up | carry/hold (an object), put down, pull (an object) |
| open (e.g., a window, a car door) | close (e.g., a door, a box), pull (an object), push (an object) |
| paint | write, draw, touch (an object) |
| play board game | play with kids, play with pets, touch (an object) |
| press | push (an object), touch (an object), turn (e.g., a screwdriver) |
| pull (an object) | push (an object), lift/pick up, carry/hold (an object) |
| put down | lift/pick up, carry/hold (an object), throw |
| shovel | dig, lift/pick up, pull (an object) |
| text on/look at a cellphone | work on a computer, answer phone, read |
| throw | catch (an object), lift/pick up, carry/hold (an object) |
| touch (an object) | point to (an object), carry/hold (an object), press |
| fight/hit (a person) | martial art, kick (a person), push (another person) |
| give/serve (an object) to (a person) | take (an object) from (a person), hand shake, carry/hold (an object) |
| grab (a person) | hug (a person), lift (a person), push (another person) |
| hand wave | hand clap, hand shake, point to (an object) |
| listen to (a person) | watch (a person), talk to (e.g., self, a person, a group), answer phone |
| push (another person) | fight/hit (a person), kick (a person), grab (a person) |
| sing to (e.g., self, a person, a group) | talk to (e.g., self, a person, a group), listen (e.g., to music), play musical instrument |
| take (an object) from (a person) | give/serve (an object) to (a person), carry/hold (an object), grab (a person) |
| talk to (e.g., self, a person, a group) | listen to (a person), watch (a person), sing to (e.g., self, a person, a group) |
| watch (a person) | listen to (a person), talk to (e.g., self, a person, a group), take a photo |

### A.7.2 THE DESCRIPTOR

The **Descriptor** module in FineAgent generates semantically meaningful captions from raw visual inputs, serving as a bridge between perceptual data and language-based reasoning. Given an image frame, the module produces a concise textual summary that captures the core scene content.

We use a Qwen2.5-VL (7B) accessed via the VLLM framework. Each frame is processed in isolation, where the model is prompted with the general instruction: *"Briefly describe this image."* This prompt is applied uniformly across all inputs to produce captions for each frame.

We intentionally omit prompts that elicit detailed scene annotations through highly specific prompts (e.g., descriptions of body posture or social interaction). This design choice is driven by a desire for *generalizability*: fine-grained prompts often induce brittle behaviors that are tightly coupled to training distributions, whereas general prompts yield more transferable representations and are more model-strength agnostic.

### A.7.3 MODEL EVALUATION ON FINEBENCH

To evaluate the performance of VLMs, we constructed a dedicated evaluation pipeline tailored to the characteristics of our proposed dataset, **FineBench** into the VLMEvalkit Duan et al. (2024)

framework. Each example in FineBench consists of a stream of temporally ordered frames, optionally accompanied by spatial cues (bounding boxes in the case of FineAgent), and a multiple-choice question requiring nuanced temporal reasoning.

**Streaming vs. Windowed Mode.** We support two data presentation paradigms. In *streaming mode*, models are presented with a causal sequence of frames leading up to a target moment, simulating a real-time inference scenario. In contrast, the *windowed mode* provides a symmetric temporal window centered on the query frame, useful for evaluating models under less temporally constrained settings. In early experiments, we found that the windowed mode slightly outperforms the streaming mode (55.6 vs 55.5 for mPlug-Owl3), therefore, we stick with the former in this paper. However, both pipelines are supported and are open-sourced to allow other researchers to experiment with them. They also have similar performance and can be interchanged.

**Caption and Spatial Grounding.** When captions are available and enabled, they are extracted using timestamp alignment and included as additional input. Furthermore, if bounding boxes are annotated, models are prompted to attend specifically to the corresponding visual region in the target frame, supporting evaluations on spatial reasoning.

**Prompt Construction.** To probe model understanding effectively, we construct prompts that emulate a multimodal dialogue. Each prompt comprises a sequence of visual and textual components. Visual context is provided via a series of temporally ordered image frames extracted around the question timestamp. These are delivered either as a sliding buffer of preceding frames (streaming mode), or a centered clip with symmetric context (window mode). The number of frames (typically 8) and their stride are configurable.

Textual prompts follow a structured template designed to focus model attention on the correct frame. For example, in streaming mode (default), the prompt reads:

> These are the frames of a video stream. Select the best answer to the following multiple-choice question based on the most recent frame (frame $t$ of $N$). Respond with only the letter (A, B, C, or D) of the correct option.

In the presence of captions, an additional block of dialogue text is inserted before the question prompt, reflecting the temporal alignment between frame timestamps and captions.

Furthermore for spatial grounding, we optionally append bounding box coordinates corresponding to relevant regions in the final frame:

> Pay attention to the region within the bounding box coordinates [x_min, y_min, x_max, y_max] in the target frame.

The final part of each prompt contains a clearly formatted multiple-choice question, e.g.:

> Question: What is the man doing in the last frame?
> A. Walking out the door
> B. Sitting on the couch
> C. Pouring a drink
> D. Looking at the camera
> Answer:

**Evaluation Protocol.** Same as other Video MCQs in VLMEvalkit (using exact matching or rule-based to extract the correct letter).

Overall, this setup ensures that models are assessed not only for answer accuracy, but also for their ability to integrate multimodal cues under temporal constraints, which is the core challenge posed by FineBench.

### A.7.4 QUALITATIVE RESULTS AND ANALYSIS

To provide a more nuanced understanding of model performance, common challenges, and the impact of FineAgent, we present qualitative examples from FineBench in Figure 7 and Figure 8.

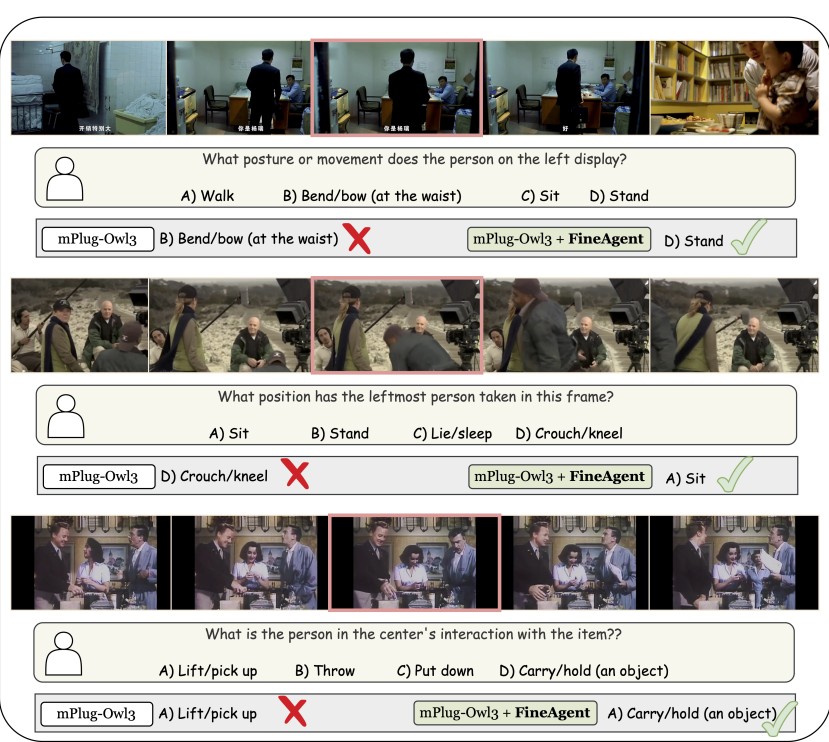

Figure 7: Qualitative examples demonstrating the improvements provided by FineAgent when applied to the mPlug-Owl3 model. The image highlights instances where the base model struggles with fine-grained distinctions in posture (e.g., distinguishing "Stand" from "Bend/bow"), position (e.g., "Sit" from "Crouch/kneel"), or nuanced object interactions (e.g., "Carry/hold" vs. "Lift/pick up"). With FineAgent's enhanced spatial grounding and contextual descriptions, the model often corrects these initial misjudgments, leading to accurate predictions.

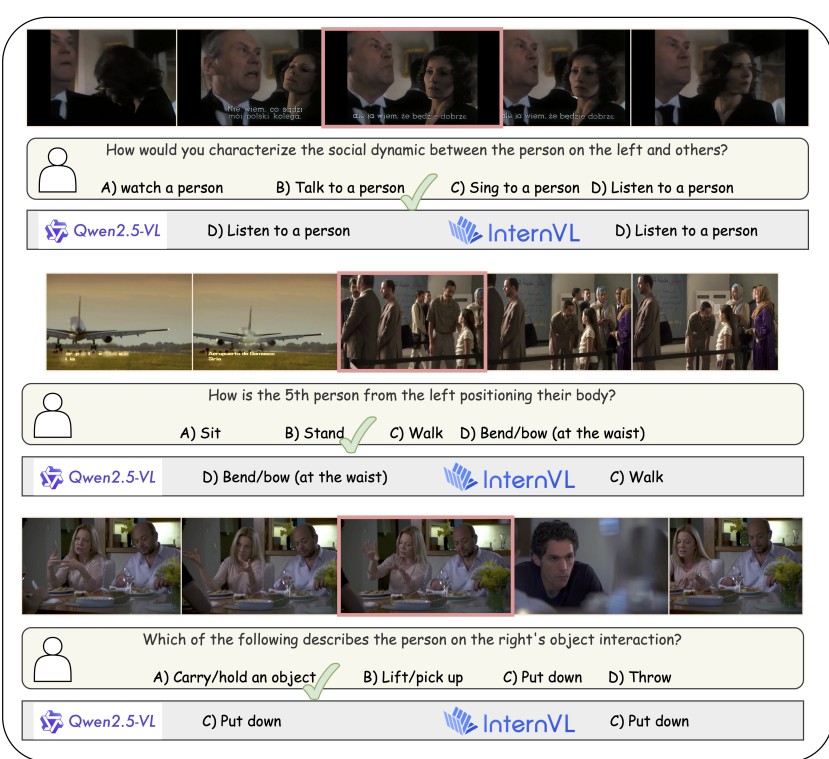

Figure 8: Qualitative examples comparing Qwen2.5-VL and InternVL on FineBench. The image illustrates scenarios including: (Top) incorrect identification of social dynamics ("Listen to a person") by both models; (Center) a common failure case where both models struggle with complex spatial referencing and posture identification in a crowded scene (e.g., "5th person from the left"), resulting in incorrect action predictions ("Bend/bow" or "Walk" instead of likely "Stand"); and (Bottom) an instance of incorrect object interaction identification ("Put down") by both models. These examples showcase the persistent challenges for VLMs in fine-grained understanding.

Figure 7 showcases scenarios where the base VLM (mPlug-Owl3) initially makes errors in identifying subtle human postures, positions, or the nature of object interactions. With the assistance of FineAgent, which provides enhanced spatial localization and descriptive context, the model is often able to correct its initial misjudgment. For instance, FineAgent helps distinguish between "Bend/bow" and "Stand," aids in differentiating "Sit" from "Crouch/kneel," and assists in understanding the continuous nature of an action ("Carry/hold") versus its initiation ("Lift/pick up"). These examples, consolidated in the figure, underscore the value of the Localizer and Descriptor components in FineAgent for tackling fine-grained tasks.

Figure 8 presents a comparison between two open-source models, Qwen2.5-VL (7B) and InternVL-2.5 (7B). The figure highlights a common failure mode: when faced with a question requiring precise identification and analysis of a specific individual in a crowded scene (e.g., "the 5th person from the left"), both models struggle, selecting incorrect actions. This points to the persistent challenges in spatial reasoning and subject disambiguation that FineBench is designed to test, and which FineAgent aims to alleviate.

### A.7.5 BROADER IMPACTS

The advancements presented in this paper, through the FineBench benchmark and the FineAgent framework, have the potential for significant broader impact. By enabling machines to achieve a more nuanced understanding of human actions and interactions, this research can contribute to the development of more capable and intuitive AI systems across various domains. Positive applications include enhanced assistive technologies for individuals with disabilities, more effective healthcare monitoring, improved human-robot collaboration, and more sophisticated content analysis tools. However, the capacity for detailed human behavior analysis also raises important ethical considerations. The potential for misuse in surveillance, the risk of perpetuating or amplifying biases present in training data, and the implications for privacy necessitate careful consideration and the development of robust ethical guidelines and safeguards. We believe that fostering research in fine-grained understanding, while being mindful of these challenges, is crucial for creating AI systems that can safely and beneficially integrate into human society.

### A.7.6 LIMITATIONS

While FineBench provides a valuable resource for assessing fine-grained human-centric video understanding and FineAgent offers a promising direction for enhancing VLMs, we acknowledge several limitations that also point towards avenues for future research.

**Limitations of FineBench** The FineBench benchmark, despite its scale and density, has limitations primarily stemming from its reliance on the AVA v2.2 dataset, which may introduce biases from cinematographic content and define the scope of "fine-grained" actions. The use of predefined templates for question and distractor generation might not fully capture natural linguistic diversity and could lead to models exploiting patterns, while the multiple-choice format primarily tests discriminative rather than generative understanding. Although featuring long videos, the 64 unique video sources might also limit broad generalization across diverse real-world scenarios compared to datasets with a wider array of distinct video origins.

**Limitations of FineAgent** The proposed FineAgent framework, while effective, presents certain limitations inherent to its modular design. These include the potential for error propagation from its Localizer or Descriptor components to the main VLM, and an increase in computational overhead and latency due to the sequential processing of these modules. The overall performance of FineAgent is also dependent on the capabilities of the underlying foundation models used for localization and captioning, and its effectiveness can be sensitive to prompt engineering and the strategy chosen for integrating auxiliary information. Lastly, FineAgent is specifically tailored to improve spatial reasoning and nuanced human action interpretation, and may not address other potential weaknesses in VLMs, such as very long-range temporal reasoning or abstract conceptual understanding.

**Future work** could focus on expanding FineBench with more diverse video sources and human-authored questions to mitigate template effects.

### A.8 LLM USAGE DISCLOSURE

In preparing this work, we made limited use of Large Language Models (LLMs), specifically OpenAI's ChatGPT, to assist with grammar checking, style refinement, and minor restructuring of text. All technical content, ideas, analyses, experimental designs, and conclusions were conceived and validated by the authors. The LLM was not used to generate novel research insights, create data, design experiments, or perform evaluations. The authors take full responsibility for the accuracy and integrity of the final manuscript.

#### A.8.1 LICENCE

AVA Gu et al. (2018) is licensed under CC BY 4.0, and our annotations are under the MIT Licence.

