# OpenReview forum: "FineBench: Benchmarking and Enhancing Vision-Language Models for Fine-grained Human Activity Understanding"
_ICLR.cc/2026/Conference — ICLR 2026 Conference Withdrawn Submission_

### Official Review · Reviewer_yHHT · 2025-10-31

**Soundness:** 2
**Presentation:** 3
**Contribution:** 2
**Rating:** 2
**Confidence:** 4

**Summary:**

The paper introduces FineBench, a human-centric video question answering (VQA) benchmark specifically designed to assess fine-grained understanding, comprising of 199,420 multiple-choice QA pairs densely annotated across 64 long-form videos (15 minutes each), focusing on detailed person movement, person interaction, and object manipulation, including compositional actions. The paper also proposes FineAgent, a modular framework that enhances VLMs by leveraging a Localizer and a Descriptor.

**Strengths:**

1. The paper is well-written and motivated, especially I agree that the problem of rigorously benchmarking fine-grained information is crucial and timely.

2. The paper evaluates many models and also proposes a method, FineAgent to improve performance, and conducts many ablations to localise the source of improvement.

**Weaknesses:**

See Questions

**Questions:**

1. While the key contribution of the paper is a new benchmark, I find the details presented in Section 3.2 regarding the dataset creation process to be severely lacking. For example, the authors mention “The primary strategy involves selecting actions that are semantically similar to the correct answer, based on a predefined similarity mapping”. What is the similarity mapping? How are the semantic similarities computed? Also, the authors mentioned that they perform subject disambiguation, is this done automatically or using manual annotations? If its the latter, do the authors conduct an inter-annotator agreement to ensure things are correctly labeled, etc? I would urge the authors to provide more details here, especially since many people might not be familiar with the vase AVA dataset.

2. The authors claim that existing benchmarks often lack a specific focus on fine-grained human-centric actions. However, I find this claim not to be empirically substantiated in the paper. In my anecdotal experiences, while other benchmarks like VideoMME have not been specifically curated for human interactions, by virtue of many videos featuring humans and their interactions they contain such questions. Can the authors empirically consolidate this point via trying to quantify the proportion of fine-grained human related questions in existing benchmarks? I think since this is one of the main novel contributions, it's important to clearly establish this contribution.

3. What is the rationale behind curating a benchmark with ~200k QAs, especially since we see that performance trends are quite similar on the ~20k benchmark as well. Since the overall question categories are not overly many, each must contain many QAs, and probably do not need so many samples to bring out discriminative power.

4. In Figure-3c), we see that the performance decreases as the number of frames are increased which is very counterintuitive to me. Especially for a finegrained understanding benchmark one would expect that performance improves as more visual information is provided. In general, improved performance with increased frame rate is considered a desirable benchmark property as it denotes the benchmark requires integrating information over different frames and is not biased towards only a singular or few frames, etc.

5. Can you compare more scaffoldings such as VideoAgent as baselines against FineAgent?

---

### Official Review · Reviewer_CAsR · 2025-10-31

**Soundness:** 3
**Presentation:** 3
**Contribution:** 3
**Rating:** 8
**Confidence:** 2

**Summary:**

The authors introduce FineBench, a dense, human-centric video VQA benchmark aimed at testing fine-grained understanding of human actions and interactions. They show that the current open VLMs struggle on these skills and also propose FineAgent, a framework that improves VLM fine-grained performance.

**Strengths:**

Novel benchmark for fine-grained video understanding. FineBench combines large scale (199,420 QA over 64 long videos) with both dense temporal and spatial grounding.

Good evaluation. The paper benchmarks a wide range of proprietary and open VLMs and surfaces concrete failure modes—accuracy drops in multi-person scenes and lower performance on Person Movement/Interaction vs. Object Manipulation

FineAgent. The modular Localizer+Descriptor add-on is well-motivated and yields consistent gains across models

**Weaknesses:**

Limited diversity and generalization scope. FineBench, while dense, is built from only 64 long videos. This narrow domain risks overfitting to cinematic, Western-style scenes and may not generalize to other video domains (e.g., egocentric, sports, social media)

**Questions:**

In figure three, it is hard to see how accuracy drops as the number of people increases.something like c plot would be much better,

---

### Official Review · Reviewer_XCgd · 2025-11-01

**Soundness:** 3
**Presentation:** 3
**Contribution:** 2
**Rating:** 4
**Confidence:** 5

**Summary:**

The paper introduces FineBench, a new benchmark designed to evaluate fine-grained, human-centric video understanding. The paper conducts a comprehensive evaluation of state-of-the-art VLMs. The paper proposes FineAgent, a modular framework leveraging spatial grounding and contextual captioning, demonstrating its effectiveness in improving the fine-grained video understanding.

**Strengths:**

1. Compared with previous benchmarks, the proposed FineBench contains larger number of QA pairs, with spatial grounding and temporal grounding in a human-centric setting.
2. It conducts thorough evaluation on current open and close -sourced models.
3. The proposed FineAgent works well several open-sourced models.

**Weaknesses:**

1. The idea of benchmarking fine-grained reasoning is not conceptually new. Several existing works already target similar objectives.
2. The taxonomy of "fine-grained reasoning" mixes low-level visual discrimination with higher-level temporal and spatial reasoning under one umbrella. The evaluation interpretation becomes vague, it is unclear whether model errors reflect poor perception, grounding or reasoning, making the proposed benchmark unreliable.
3. The paper lacks a comprehensive analysis of how current models failed in each type of QA. (e.g., failure type breakdown, visual attention maps, per-category robustness) And lack of visual examples.

**Questions:**

1. How do you identify which person in the video is the "person" in the question referring to?
2. How do you review the annotation quality of the benchmark? Do you have audio information? In Fig1(b), it is even hard for humans to understand which person is talking at the moment.

---

### Official Review · Reviewer_CLti · 2025-11-10

**Soundness:** 3
**Presentation:** 3
**Contribution:** 3
**Rating:** 6
**Confidence:** 4

**Summary:**

The paper presents FineBench, a large-scale benchmark for fine-grained human-centric video question answering (VQA), containing about 200k QA pairs densely annotated across 64 long-form videos. It aims to evaluate models’ ability to reason about subtle human actions, movements, and interactions. The authors also introduce FineAgent, a modular enhancement that combines a Localizer for spatial grounding and a Descriptor for contextual captioning. Experiments demonstrate that FineAgent improves multiple open-source vision-language models (VLMs) on FineBench.

**Strengths:**

- FineBench is a well-motivated benchmark that significantly extends existing VQA datasets by emphasizing dense spatial and temporal grounding in human-centric videos.
- The empirical evaluation is thorough, covering both open and proprietary models with meaningful analyses of weaknesses (e.g., multi-person disambiguation, fine motion reasoning).
- FineAgent is a practical framework that yields consistent improvements, and its modularity makes it easy to adapt to different models.
- Figures and tables are clear and well connected to the text. The dataset design and annotation process are transparent and reproducible.

**Weaknesses:**

- The paper lacks discussion and comparison with several recent fine-grained video benchmarks (e.g., MotionBench, VER-Bench, Finer), which weakens the positioning of FineBench’s novelty.
- The dataset construction, though detailed, relies heavily on templated questions and lacks formal mathematical specification for distractor generation, which could limit reproducibility.
- Experiments on downstream or transfer tasks are missing, making it unclear whether FineBench improvements generalize to real-world applications.
- The scalability and computational cost of FineAgent are only briefly mentioned; more quantitative profiling would strengthen the paper.
- Some aspects of the contribution feel incremental rather than conceptually novel, as the work builds on existing datasets and modular reasoning designs.

**Questions:**

1. Can the authors provide formal notation or pseudocode for distractor generation and spatial referencing?
2. How does FineBench differ experimentally from MotionBench or VER-Bench?
3. Have the authors tested FineAgent with different Descriptor models to assess its generality?

---

### Note · Authors · 2025-11-12

I have read and agree with the venue's withdrawal policy on behalf of myself and my co-authors.